# Diversity impact on organizational performance: Moderating and mediating role of diversity beliefs and leadership expertise

**Jamshid Ali Turi[1], Sudhaishna Khastoori[2], Shahryar Sorooshian[3]\*, Nadine Campbell[4]**

1 Department of Management Studies, Bahria Business School, Bahria University, Islamabad, Pakistan,
2 Department of Management Sciences, SZABIST, Larkana, Pakistan, 3 Department of Business Administration, University of Gothenburg, Gothenburg, Sweden, 4 Business school, Western Sydney University, Sydney, Australia

\* shahryar.sorooshian@gu.se

**Data Availability Statement:** The data has been sent to the SZABIST center of research in Shaheed Zulfikar Ali Bhutto Institute of Science and Technology, 79 Clifton Road, Karachi 75600,

## Abstract

The current research examines the impact of four independent diversity variables, gender, age, educational background, and ethnicity, on the moderating role of diversity beliefs and the mediating role of leadership expertise to measure organisational performance in Pakistan. A self-administered questionnaire using a 6-point Likert scale approach was adopted to collect the responses from 176 employees. Quantitative analysis was done using SPSS, and SMART-PLS3 were used for was used to comprehend the objectives of the research. The findings indicate that age diversity, diversity beliefs, and leadership expertise have a statistically significant impact on organisational performance. Moreover, moderating variable diversity belief did not affect organisational performance, but leadership expertise plays a significant mediating role in organisational performance. Our study provides critical theoretical contributions to research diversity and organisational performance in Pakistan and examines the impact of workforce diversity on organisational performance with leadership expertise as mediator and diversity beliefs as a moderator.

## 1. Introduction

Diversity has many meanings, applications, and implications. Some organisations see it as an asset from which innovation and competitive advantages can springboard, while others see it as a hindrance, constrain, and biases. Traditionally, diversity included religion, language, age, gender, ethnicity, education, cultural and personality orientation [1]. Today, the concept of diversity has evolved to encompass strategic targets to improve organisational performance and effectiveness [2]. Therefore, organisations promote workforce diversity to bolster organisational performance [3]. However, many studies suggest that diversity exists in different forms with different intensities. If not managed properly, it has the potential to harm morale, intensify turnover and result in substantial communication problems.

The lack of diversity training and understanding of diversity beliefs, especially in developing countries with rigid social and cultural bonds, leads to organisational bias. To overcome

Pakistan. To obtain the archive, should use the main author's name followed by the year 2021, "Ali Turi / Khastoori - 2021. Data set, the contact information for the SZABIST center of research is: Tel: (021) 358-21538-42 (EXT # 407) Fax: (021) 35830446 Email: info@szabist.edu.pk.

**Funding:** The author(s) received no specific funding for this work.

**Competing interests:** The authors have declared that no competing interests exist.

these organisational biases, E-Vahdati et al. [4] recommended that firms should emphasise corporate governance, accountability, ethics, trust, and diversity. Moreover, organisations also need diversity for rational decision-making and promoting a conducive environment, where everyone's beliefs are respected, leading to employees self-reflecting on the positive benefits [5,6]. However, if workforce diversity is mismanaged, this could lead to emotional conflicts, perceived organisational politics, miscommunication, power struggle, and higher employee turnover. As a result, having a diverse workforce would become an inhibitor for organisational development [7,8].

Muhammad Ali Jinnah, the founder of Pakistan, believed that diversity management involves four key concepts. One is democratisation which would guarantee cooperation amongst its citizens. Two, consistent social equity and equivalence through egalitarian Islamic values. Three, stringent laws with no room for bias or discrimination. Four, protectionism for minorities, women, and other disadvantaged groups [9]. Despite this, Pakistan is among the lowest-ranked diverse countries in the world. It ranked in the 22nd percentile for gender diversity and female economic activity in emerging economies due to its religious and cultural norms. Additionally, Pakistan's sectoral diversity falls in the bottom five [10].

Previous studies on diversity focused on culture and ethnicity, but elements such as age, gender, and education have not been fully explored. Therefore, there is a need to examine different elements of diversity in different settings to understand its applications and managerial implications for sustainable organisational performance [11–13]. However, the subjective nature of diversity has left many practitioners ill-equipped to manage diversity effectively or determine which components play a role in diversity management and diversity-related issues [14].

The contradictory research results on diversity need to be further examined to increase our comprehension and better explain this phenomenon. Previous research has considered various diversity dimensions to identify their impact on organisational performance. For example, García-Granero et al. [15] and Georgakakis [16] explored the relationship between top management team functional diversity and the firm's performance with the moderating role of top management (CEO) attributes. Other studies have used negative descriptors such as discrimination and racial prejudice to explore diversity.

However, no studies have examined the projectized environments or considered the role of leadership expertise and diversity beliefs. This research's main queries are to determine how leadership expertise adds to organisational performance, value diversity beliefs, and organisational performance? Therefore, our contribution to the diversity literature will help us better understand and assess the impact of diversity on organizational performance by examining leadership expertise as a mediating variable and determining the extent to which diversity and organizational performance are related, using diversity beliefs as a moderating variable within Pakistan.

## 2. Literature review and hypotheses

Diversity is considering, recognising, and respecting others' opinions and differences irrespective of their culture, gender, age, social status, race, physical capability, and so on [7,17]. It is used to find opportunities, face challenges, and explore new avenues [18]. Furthermore, diversity can be used to enhance knowledge and skill levels, help to understand behaviour, conflicts and fill the gaps within the organisation [7,19]. While there are many facets to diversity, this research aims to look more especially at gender, age, ethnicity, and educational diversity.

### 2.1 Gender diversity

Gender diversity represents the gender identities of men and women. It describes the emotional difference and experience publicly and culturally attached to men and women within

any firm [20]. Research has found that a moderate level of gender diversity boosts the competitive edge, whereas greater levels of gender diversity reduce organizational performance. Other studies have shown that organisational success depends upon gender equality and equity [21,22];. Although western organisations have been moving closer to gender equality, Pakistan is way behind [21]. The gender-oriented inequities within the Pakistani workplace are reinforced by personal biases and stereotypes, referring that the status of men is perceived as superior to women. Many organisations prefer hiring male employees because they perceive men as better performers [23].

## 2.2 Age diversity

*Age diversity* is the ability of an organisation to accept different age groups. The business environment can only grow and succeed when various age groups within an organisation come with diverse experiences [24–26]. Recently, age diversity issues have gained significance because professionals are choosing to work past retirement age, and young adults are working part-timers while completing their studies [27–29]. Many organisations are welcoming this trend because they need skilled employees with experience and young talent with an innovative mindset for new ventures better organisational performance [30,31]. However, In Pakistan, young people face more discrimination in the labour market than old workers [32], as cultural norms are founded on respect for their elders.

## 2.3 Ethnic diversity

Ethnic diversity refers to differences in religion, language, and cultural background. Employees from different backgrounds working in the same organisation represent different lifestyles, cultures, beliefs, and skills that can improve strategic decisions [14]. Due to these perceived attributes and globalisation, organisations are focusing on multiplicity diversity building, but many companies struggle to produce and implement policies that reduce ethnic discrimination, which negatively impacts organisational performance [32–35]. Pakistani laws espouse that all citizens are equal irrespective of their religion, language, gender, or caste, but for minorities in Pakistan, this is a farfetched dream. According to EEOC data, ethnic diversity violations cost companies $112.7 million per annum due to ethnic diversity violations [3].

## 2.4 Educational diversity

Educational diversity denotes differences in knowledge, training, skills, experience, and qualification [18,36]. Some organisations refuse to employ highly qualified workers because they do not believe highly educated individuals are better performers, while others see employees with less education, skills, and training underperform [22]. The lowest level of education affects the earnings of rural workers in Pakistan, but old earners who receive more education earn more in urban areas. Organisations use educational diversity to have a mix of soft and hard-tech skills [37], and employees consider having educational diversity to significantly increase their ability in obtaining desirable jobs [38,39]. Age, gender, ethnicity, and educational diversity add to the synergetic pragmatism of the projects and organisation [30,40]. These findings lead us to the stance that *H1*: *Diversity has a significant positive impact on project performance.*

## 2.5 Leadership expertise

*Leadership expertise* plays a crucial role in organisational performance, as it creates new directions, new philosophies, optimism, boost enthusiasm and cooperation among employees, and devises appropriate visions and strategies. Furthermore, leadership expertise considers

diversity an organisational strength and promotes inclusion and diversity using various leadership styles as one leadership style may not work in diverse teams. The leader-member exchange (LMX) theory explains this approach best. It is a relationship-based approach with a dyadic relationship between the leader and their employees [41].

According to LMX [41], a leader uses a specific leadership style for each team member based on their mindset. The leaders share more knowledge and information, delegate responsibilities, and encourage participation in decision-making with some members and not others. LXM theory allows leaders to develop in-groups and spend more resources on the members they expect to perform better. This relationship between a leader and members gradually develops and reaches a high degree of dependence, mutual trust, and support. As a result, productivity increases. That eventually enhances employee retention, loyalty, and sustainable organisational growth.

Previous results maintain that effective diversity management at the workplace adds to both organisational and organisational performance [7,40]. Diversity, which has become an integral part of every organisation and project in this unified world, needs better leadership expertise to manage it at the micro and macro levels [34,42]. Research supports that a leader's expertise, i.e., leading employees with respect regardless of their caste and creed, leading them with self-assurance, positively shaping their behaviour, results in enhanced employee performance, which eventually reflects increased organisational performance [43]. The findings lead us to *H2*: *Diversity with leadership expertise has a positive impact on organisational performance.*

## 2.6 Role of diversity beliefs as a moderating variable

Diversity beliefs mean understanding that everyone is unique, and there is a need to recognise individual differences. These differences include race, ethnicity, gender, sexual orientation, socio-economic status, age, physical abilities, religious beliefs, political beliefs, or other ideologies [11]. Today, globalisation is one of the driving forces of diversity within organisations. However, accommodating diversity beliefs in terms of spiritual, cultural, and political views sometimes challenges a diverse organisation [12,25]. Staff needs to be reminded that they should not impose their opinions on others as their personal and ethnic beliefs are independent of their work obligations [27,44]. The employment practices linked with unbiased diversity beliefs can lead to constructive organisational results [11,26].

These diversity beliefs can be polarised perceptions or preferences towards homogeneity or heterogeneity [7,17]. A leader's diversity beliefs may be one of the factors influencing organisational performance. Manoharan and Singal [42] found diversity positively affects organisational performance when supported by positive beliefs and values. Kundu and Mor [45] concluded that a generally positive view of workforce diversity could positively impact organisational and new venture (project) performance. Additionally, the perception of employees about workforce diversity is positively linked with organisational performance [46], and employees perceive their organisation more favourably when diversity management is perceived as positive [18]. However, due to organisational variations and cultural settings, diversity needs to be managed differently [14,47]. As such, we hypothesise that *H3*: *Diversity beliefs moderates the relationship between leadership expertise and organisational performance.*

Furthermore, organisations bring people from different cultures to boost creativity, knowledge, and rational problem-solving approaches. Consequently, the leaders in this 21st century have become highly alarmed with diversity management in organisations [48]. It is believed that diversity at the workplace positively impacts organisational performance, and the leadership expertise mediates this relationship. According to prior research [8,49], organisational leaders play a vital role in forming and promoting the workplace culture, free of prejudice and

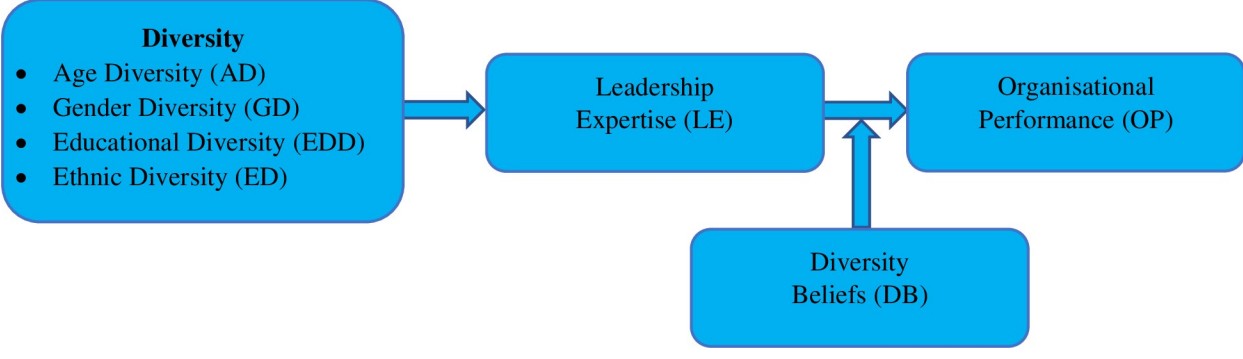

**Fig 1. Conceptual model.**

personal biases. The workforce mainly follows leaders to set the perspective wherein they would work in an organisational setting. Thus, forming such an environment that imitates respect, ethical behaviour, understanding, and encouraging cross-cultural values improves organisational performance. However, this relationship is moderated by the diversity beliefs. Everyone in the organisation does not hold the same values and beliefs. Still, a true leader who can determine the varied beliefs of employees and manage diversity in a way that is convincing for each team member can help organisations reach new heights [50]. The research findings lead to the hypothesis that *H4: Diversity significantly impacts organisational performance with the mediation of leadership expertise and moderation of diversity beliefs.*

The Conceptual Model (Fig 1) was developed based on the relationship between four dimensions of diversity most relevant to the Pakistani context, the leadership expertise, diversity beliefs, and organisational performance. This conceptual framework indicates the impact of workforce diversity on organisational performance in the presence of leadership expertise as mediating variable and diversity beliefs as moderating variable in the services sector and projectized organisations in Pakistan's major cities.

## 3. Methodology

A quantitative approach using a correlational study was undertaken to determine the extent of a relationship constructs under investigation. A structured questionnaire was adopted from previous studies [51,52] to collect primary data using a survey, keeping in mind the objectives of the studies. The study used a 6-point Likert scale for grading the responses with the scale (1 = strongly disagree, 2 = disagree, 3 = partially disagree, 4 = partially agree, 5 = agree, 6 = strongly agree). The target population of the study was the project management professionals, working in the major cities of Pakistan. These cities were selected because many of the national and international developmental projects take place here. Organizations were selected from the services sector. The questionnaires were self-administered.

Additionally, a muti-level sampling procedure was adopted to make the respondent selection process more accurate and precise. In the first phase, stratified random sampling was applied to select the strata of the potential respondents. In the second phase, the quota sampling technique was applied to select the qualifying organizations, and in the third phase, convenient sampling was used to collect data. A total of 550 questionnaires were distributed, and 482 were returned. Questionnaires were assessed and screened for completeness. A total of 17 questionnaires were discarded as more than 10% of the values were missing. A further 12 were removed because of outliers. The remaining 451 were analysed using SPSS and Smart PLS.

## 4. Results and findings

### 4.1 Participant demographics

Table 1 contains the demographic details of the respondents. Among 176 respondents, 97 were male, and 79 were female. Most of the respondents were aged 30–35, had more than 5years' experience working for their organisation, and held a bachelor's degree or higher. This indicates that the participants were well educated and possessed sufficient skills and knowledge to answer all the survey questions proficiently.

### 4.2 Instrument validity

Table 2 indicates the loading factors for all the items are in the acceptable range of greater than 0.70. The average variance extracted (AVE) falls between 0.612–0.678 for the constructs, indicating a high-reliability level. Moreover, the composite reliability (CR) values range from 0.862 to 0.947 and are highly consistent and satisfy the convergent validity criteria. Furthermore, predictive accuracy, effect size, and predictive relevance were conducted for the goodness of fit, and their values fell in an acceptable range.

### 4.3 Discriminant validity: Fornell-Larcker Criterion

Discriminant validity of the constructs was checked using Fornell-Larcker Criterion. Discriminant validity confirms correlation among constructs if the values do not exceed 0.85 and the square root of AVEs is greater than the correlation of other constructs. Table 3 maintains that all values are less than 0.85, and their square root of AVEs was greater than their constructs' off-diagonal values. These details satisfy the discriminant validity requirements.

### 4.4 Discriminant validity: HTMT Criterion

HTMT refers to the average of the correlations of indicators between different constructs relative to the average of the correlations of indicators within the same construct. It measures the discriminant validity between the construct of the instrument. While conservative cut-off values are 0.9 is advocated a more stringent ratio of 0.85 as it offers the best criterion compared to all other methods of assessing discriminant validity [53]. Thus, any inter-construct ratio greater than 0.85 would be considered as having poor discriminant validity. The HTMT ratios obtained in this study, as shown in Table 3, indicate no discriminant validity problems between the constructs.

### 4.5 Hypothesis testing

The path estimation or hypothetical relations was performed to observe the significant relationship in the inner path model. The entire hypothetical path in the framework was examined

**Table 1. Demographics of the respondents.**

| Age | Frequency | Percent | Service | Frequency | Percent |
|---|---|---|---|---|---|
| 30–35 | 214 | 47.45 | 1–5 years | 182 | 40.35 |
| 36–40 | 140 | 31.04 | 5–10 years | 160 | 35.47 |
| 41-above | 97 | 21.50 | >10 years | 109 | 24.16 |
| Education | Frequency | Percent | Gender | Frequency | Percent |
| Bachelor | 244 | 54.10 | Male | 240 | 53.21 |
| Master | 168 | 37.25 | Female | 211 | 46.78 |
| PhD | 39 | 8.64 | | | |

**Table 2. Confirmatory factor analysis for research constructs.**

| Constructs* | Item No* | Factor Loading** | AVE | CR | Goodness of Fit Indices | | | |
|---|---|---|---|---|---|---|---|---|
| | | | | | $X^2/df$ | $Q^2$ | $R^2$ | $F^2$ |
| OL | 3 | .738 - .911 | .678 | .911 | 1.171 | 0.244 | 0.452 | 0.294 |
| AD | 4 | .754 - .926 | .671 | .932 | 2.692 | 0.171 | 0.437 | 0.202 |
| ED | 5 | .833 - .855 | .753 | .927 | 1.273 | 0.204 | 0.445 | 0.083 |
| GD | 4 | .837- .840 | .764 | .947 | 1.114 | 0.365 | 0.476 | 0.381 |
| EDD | 7 | .826 - .839 | .784 | .937 | 2.925 | 0.272 | 0.229 | 0.021 |
| LE | 4 | .744 - .840 | .612 | .862 | 1.817 | 0.293 | 0.427 | 0.201 |
| DB | 3 | .714 - .869 | .674 | .884 | 2.903 | 0.213 | 0.341 | 0.217 |

*OL = Organisational Leadership; AD = Age Diversity; ED = Ethnic Diversity; GD = Gender Diversity; EDD = Educational Diversity LE = Leadership Expertise;
DB = Diversity Beliefs.

through the regression coefficient (β). Using the PLS Bootstrap technique, the value of β was checked to observe the proposed hypotheses in the structural model. Table 4 demonstrates the path coefficient assessment result where out of 10 direct hypotheses, six were supported, and four were not supported. The supported hypotheses were significant at least at the level of 0.05, have expected positive sign directions, and consist of a path coefficient value (β) ranging from 0.181 to 0.515.

Additionally, Table 5 shows that all six direct relationships were significant as the p-value is less than 0.05 and the t-value is higher than 1.96, depicted in Fig 2. However, the other four hypotheses were unsupported because the p-value was higher than 0.05, and the t-values were less than 1.96.

In the case of moderating hypothesis, DB does not moderate the relationship between LE and OP. Therefore, it confirms that DB does not play any significant moderating role in the relationship between LE and OP.

## 4.6 Mediation hypothesis

For the mediating analysis, the bootstrapping technique was applied [54]. The mediation analysis results are presented in Table 6 and in Fig 3, where among the four mediating hypotheses, three were supported, and one was not supported. The mediating path AD -> LE -> OP, ED -> LE -> OP, and EDD -> LE -> OP was significant as p < .005 and the values of LL and UL do not have zero (0) in between, which confirmed a mediating effect. However, the other

**Table 3. Square roots of AVEs.**

| Constructs* | AD | DB | ED | EDD | GD | LE | OP |
|---|---|---|---|---|---|---|---|
| AD | 0.759 | | | | | | |
| DB | 0.593 | 0.844 | | | | | |
| ED | 0.650 | 0.543 | 0.777 | | | | |
| EDD | 0.596 | 0.591 | 0.571 | 0.758 | | | |
| GD | 0.638 | 0.412 | 0.448 | 0.575 | 0.820 | | |
| LE | 0.730 | 0.690 | 0.653 | 0.623 | 0.564 | 0.837 | |
| OP | 0.726 | 0.706 | 0.608 | 0.602 | 0.546 | 0.833 | 0.847 |

*AD = Age Diversity; DB = Diversity Beliefs; ED = Ethnic Diversity; EDD = Educational Diversity GD = Gender Diversity; LE = Leadership Expertise;
OP = Organisational Performance.

**Table 4. HTMT values.**

| Constructs | AD | DB | ED | EDD | GD | LE | OP |
|---|---|---|---|---|---|---|---|
| AD | | | | | | | |
| DB | 0.762 | | | | | | |
| ED | 0.792 | 0.634 | | | | | |
| EDD | 0.759 | 0.723 | 0.671 | | | | |
| GD | 0.806 | 0.504 | 0.512 | 0.701 | | | |
| LE | 0.834 | 0.834 | 0.739 | 0.734 | 0.657 | | |
| OP | 0.811 | 0.809 | 0.710 | 0.734 | 0.658 | 0.789 | |

*AD = Age Diversity; DB = Diversity Beliefs; ED = Ethnic Diversity; EDD = Educational Diversity GD = Gender Diversity; LE = Leadership Expertise;

OP = Organisational Performance.

mediating path GD -> LE -> OP was not significant as p < .005, and the zero (0) exists between LL and UL. In addition, among the three hypotheses, the AD -> LE -> OP path was partially mediated as the direct hypothesis was significant. However, the other two significant paths were fully mediated as their direct relationships were not significant.

## 5. Discussion

After many years of research on workplace diversity, there is considerable misperception over what diversity is. The broad definitions state that diversity seeks inclusion but does not identify the difference between social diversity where individuals of different races, ethnicity, religious beliefs, socio-economic status, language, geographical origin, gender, and/or sexual orientation bring their different knowledge, background, experience, and interest to increase organisational performance. Similarly, functional diversity where individuals with a variety of educational and training backgrounds are not examined. As a result, organisations are left confused about how to manage diversity to maximise organisational performance [55–58].

The present research provides a better understanding of the prevailing diversity scenario in Pakistan's service sector and projectized organisations. The research indicates that three diversity variables, ethnic, gender, and education, do not significantly impact organisational performance. In contrast, age diversity has a significant impact on organisational performance.

**Table 5. Direct and moderating hypothesis.**

| Hypotheses | OS | SM | SD | T | P Values | Decision |
|---|---|---|---|---|---|---|
| AD -> LE | 0.401 | 0.396 | 0.082 | 4.865 | 0.000 | Significant |
| AD -> OP | 0.181 | 0.187 | 0.091 | 1.992 | 0.007 | Significant |
| DB -> OP | 0.212 | 0.217 | 0.079 | 2.674 | 0.008 | Significant |
| ED -> LE | 0.242 | 0.247 | 0.077 | 3.165 | 0.002 | Significant |
| ED -> OP | 0.013 | 0.013 | 0.087 | 0.151 | 0.880 | Not Significant |
| EDD -> LE | 0.195 | 0.191 | 0.078 | 2.521 | 0.001 | Significant |
| EDD -> OP | 0.021 | 0.017 | 0.062 | 0.331 | 0.741 | Not Significant |
| GD -> LE | 0.087 | 0.096 | 0.089 | 0.976 | 0.330 | Not Significant |
| GD -> OP | 0.037 | 0.039 | 0.066 | 0.568 | 0.570 | Not Significant |
| LE -> OP | 0.515 | 0.508 | 0.087 | 5.952 | 0.000 | Significant |
| LE*DB -> OP | -0.005 | -0.005 | 0.027 | 0.186 | 0.853 | Not Significant |

*AD = Age Diversity; DB = Diversity Beliefs; ED = Ethnic Diversity; EDD = Educational Diversity GD = Gender Diversity; LE = Leadership Expertise;

OP = Organisational Performance.

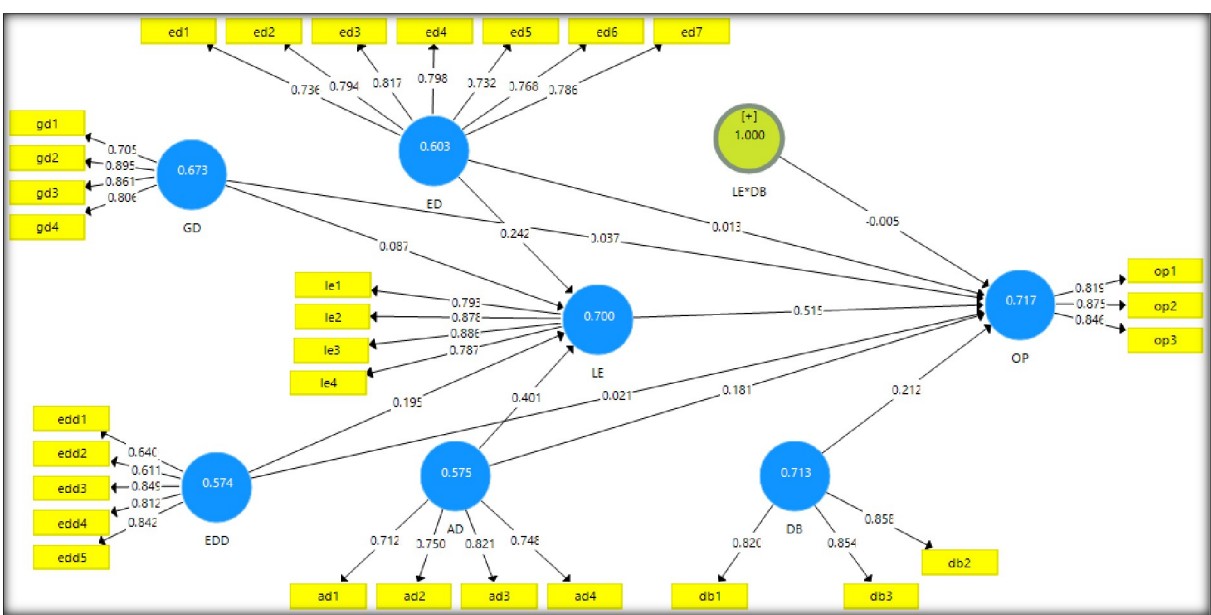

**Fig 2. PLS-Algorithm result with outer loadings and AVE.**

The moderating hypothesis indicates that diversity beliefs play no significant role in improving organisational performance. This study challenges previous findings in the literature review sections, which proclaims that diversity and diversity beliefs significantly affect organisational performance. Therefore, organisations prefer to engage the workforce with diverse social, cultural, and ethnic backgrounds, bringing multi-facet experiences, learning, tacit and explicit knowledge to the organisation, boom effectiveness, and efficiencies, face challenges, and accept future challenges. This may be due to regional and cultural factors, that diversity beliefs are not promoting organisational performance, which may be explored in the future. Moreover, this study indicates that leadership expertise plays a significant mediating role, and diversity beliefs play a significant moderating role in organisational performance.

## 5.1 Theoretical implications

Our study provides critical theoretical contributions to research diversity and organisational performance. There is a gap in the current literature on the impact of workforce diversity on organisational performance, with leadership expertise as mediating variable and diversity beliefs as moderating variable in the services sector and projectized organisations in Pakistan.

**Table 6. Mediation hypothesis.**

| Hypothesis | OS (Beta) | 95% Confidence Interval | | T | P | Decision | Mediation |
|---|---|---|---|---|---|---|---|
| | | LL | UL | | | | |
| AD -> LE -> OP | 0.206 | 0.122 | 0.345 | 3.743 | 0.000 | Significant | Partial Mediation |
| ED -> LE -> OP | 0.125 | 0.055 | 0.222 | 2.964 | 0.003 | Significant | Full Mediation |
| EDD -> LE -> OP | 0.101 | 0.025 | 0.210 | 2.226 | 0.001 | Significant | Full Mediation |
| GD -> LE -> OP | 0.045 | -0.034 | 0.150 | 0.973 | 0.331 | Not Significant | No Mediation |

*AD = Age Diversity; DB = Diversity Beliefs; ED = Ethnic Diversity; EDD = Educational Diversity GD = Gender Diversity; LE = Leadership Expertise; OP = Organisational Performance.

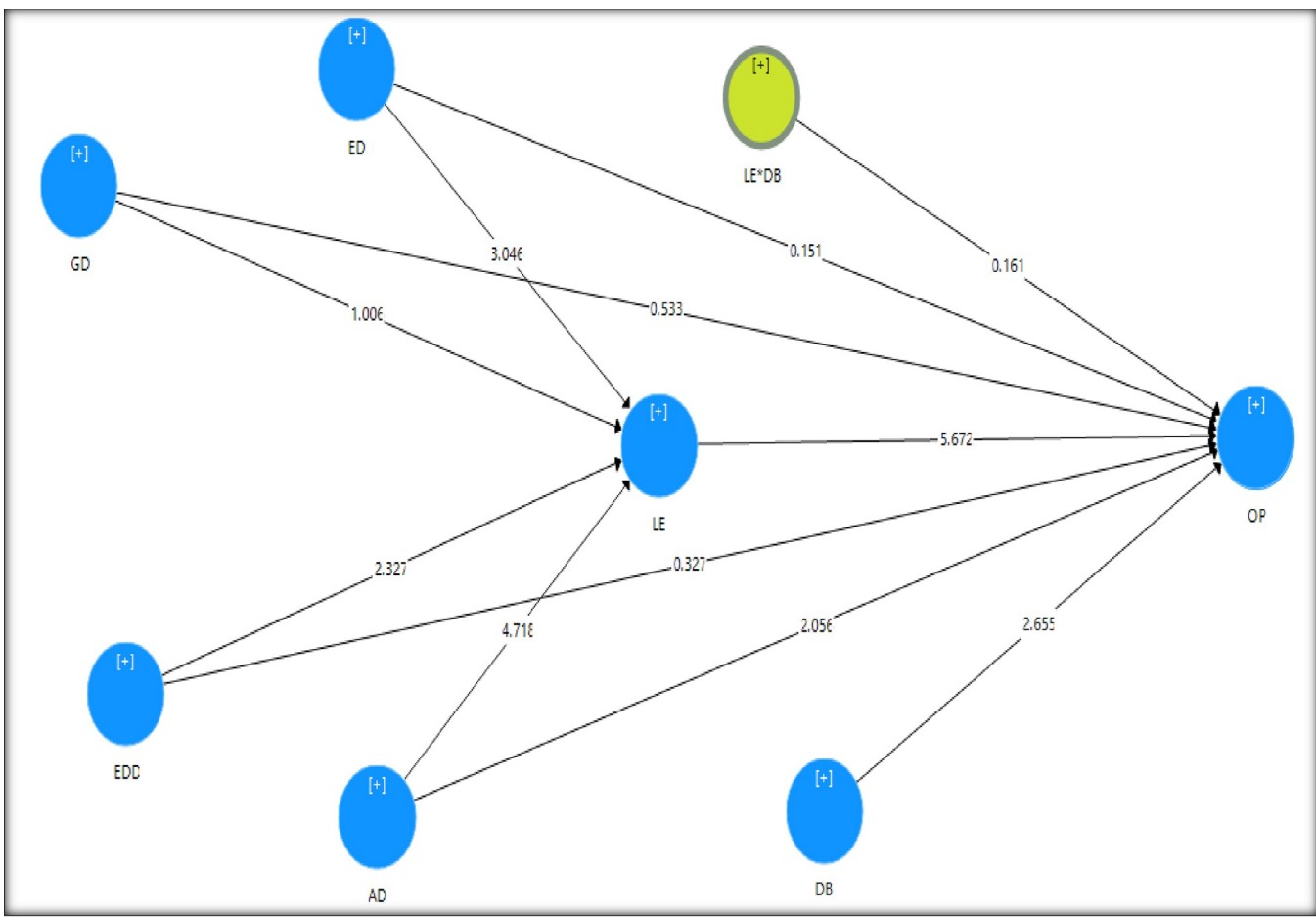

**Fig 3. Bootstrapping result with inner t-values.**

Specifically, we determined that leadership expertise mediates age, ethnicity, and educational diversity, and organisational.

Second, we contribute to research on the effective path by which diversity influences organisational performance by exploring the mediating role of leadership expertise. That is, our study not only examined that leadership expertise positively influences organisational performance. Building on these studies, our research uses leader-member exchange theory as an effective path and organisational performance as a goal. Drawing on the leader-member exchange theory, we determine that leadership expertise can impact diversity and enhance organisational performance. Our results suggest that leadership expertise is a crucial mechanism for diversity management and improving organisational performance in Pakistan.

Finally, our research explored the value of incorporating the moderator, diversity beliefs, and the mediator leadership expertise into a single theoretical model helps us better to understand the relationship between diversity and organisational performance. Our study showed that diversity beliefs do not moderate the relationship between leadership expertise and organisational performance. However, there were direct relationships between age diversity and leadership expertise, age diversity and organisational performance, diversity beliefs and organisational performance, and ethnic diversity and leadership expertise. Additionally, this study also found that there is partial and no mediation between age diversity, gender diversity, and organizational performance.

## 5.2 Practical implications

In addition to the theoretical contributions, our research informs practitioners in several ways. First, our results show that age, ethnicity, and educational diversity directly contributes to organisational performance via leadership expertise. There was also a direct relationship between age and ethnic diversity and leadership expertise. These findings emphasise the relevance of diversity management in light of globalisation.

Leaders should employ leader-member exchange procedures to help sustain organisational performance in an increasingly diverse workforce. That is, leadership styles need to change based on the mindset of the various groups within the organisation. The leaders share more knowledge and information, delegate responsibilities, and encourage participation in decision-making with some members and not others. LXM theory allows leaders to develop in-groups and spend more resources on the members they expect to perform better. However, this study added to the body of knowledge, that leadership expertise may not contribute to well managed and effective group development, due to social, religious, and cultural limitations of the locality/respondents.

## 5.3 Limitations and future research directions

This study has several limitations. First, it focused on age, gender, ethnic, and education diversity management and did not take into account other demographic diversity practices implemented within the organisations. Previous research recognises that a broad spectrum of demographic diversity influences organisational performance [55]. Future research should investigate a broader range of demographic diversity to understand better what constitutes a comprehensive approach to diversity management. Second, the research is quantitative, and its moderate response rate may limit the generalisability of the results [59]. Future research could combine qualitative and quantitative methods to leverage both structured and unstructured data to enhance the depth of insights and provide more specific practical outcomes [60]. Third, the generalisability of findings should be interpreted with caution. Every society has its own culture, norms, and social values, and previous research has identified that organisational culture may influence the findings related to diversity management [61].

## 6. Conclusions

Workplace diversity is becoming one of the most popular ways to evaluate organisational performance. Thus, conducting training and creating awareness regarding diversity will lead to value generation, better productivity, and vitality. Managing diversity at the workplace considers leveraging and respecting cultural differences in employees' competencies, ideas, and innovativeness to persuade them to contribute towards a common goal and do it in a way that gives a competitive edge to organisations. Hence, it is recommended to encourage a more diversified workforce and create awareness to increase organisational performance. In addition, this research has focused on diversity beliefs as a moderating variable. However, future research can be conducted that how leadership expertise can mediate between age and gender diversity and organizational performance. Additionally, organisational justice as a moderator between diversity dimensions and organisational performance needs to be explored. Moreover, in the current paper, the social traits of diversity have been studied, providing opportunities or gaps to study functional diversity traits in the future.

## Acknowledgments

### Ethical consent

The study was approved by the ethical committee of the SZABIST Larkana Campus. The consent was informed, and the information was collected through an approved structured questionnaire. Moreover, the authors declare that they have no conflict of interest.

## Author Contributions

**Conceptualization:** Jamshid Ali Turi, Shahryar Sorooshian.

**Data curation:** Jamshid Ali Turi, Sudhaishna Khastoori.

**Formal analysis:** Jamshid Ali Turi, Sudhaishna Khastoori.

**Methodology:** Jamshid Ali Turi.

**Project administration:** Jamshid Ali Turi.

**Supervision:** Jamshid Ali Turi.

**Validation:** Sudhaishna Khastoori.

**Visualization:** Jamshid Ali Turi.

**Writing – original draft:** Jamshid Ali Turi, Sudhaishna Khastoori.

**Writing – review & editing:** Shahryar Sorooshian, Nadine Campbell.

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
