## [Decision Letter · Decision Letter 0]

16 Dec 2021

PONE-D-21-09151Diversity Impact on
Organizational performance: Moderating and Mediating role of Diversity Beliefs and
Leadership ExpertisePLOS ONE

Dear Dr.
Sorooshian,

Thank you for submitting your manuscript to PLOS ONE. After careful consideration, we
feel that it has merit but does not fully meet PLOS ONE’s publication criteria as it
currently stands. Therefore, we invite you to submit a revised version of the
manuscript that addresses the points raised during the review process.

Please include the following items when submitting your revised
manuscript:A rebuttal letter that responds to each point raised by the academic
editor and reviewer(s). You should upload this letter as a separate file
labeled 'Response to Reviewers'.A marked-up copy of your manuscript that highlights changes made to the
original version. You should upload this as a separate file labeled
'Revised Manuscript with Track Changes'.An unmarked version of your revised paper without tracked changes. You
should upload this as a separate file labeled 'Manuscript'.

If you would like to make changes to your financial disclosure, please include your
updated statement in your cover letter. Guidelines for resubmitting your figure
files are available below the reviewer comments at the end of this letter.

We look forward to receiving your revised manuscript.

Kind regards,

José Gutiérrez-Pérez

Academic Editor

PLOS ONE

Journal Requirements:

3.  Please improve statistical reporting and refer to p-values as "p<.001" instead
of "p=.000". Our statistical reporting guidelines are available at https://journals.plos.org/plosone/s/submission-guidelines#loc-statistical-reporting 

4. Please consider changing the title so as to meet our title format requirement
(https://journals.plos.org/plosone/s/submission-guidelines). In
particular, the title should be "Specific, descriptive, concise, and comprehensible
to readers outside the field" and in this case it is not informative and specific
about your study's scope and methodology (in particular the study is observational
and based on self-reports from surveys).

5. Please provide additional details regarding participant consent. In the Methods
section, please ensure that you have specified (1) whether consent was informed and
(2) what type you obtained (for instance, written or verbal). If your study included
minors, state whether you obtained consent from parents or guardians. If the need
for consent was waived by the ethics committee, please include this information.
Furthermore, please include your ethics statement in the online submission.

7. Please note that in order to use the direct billing option the corresponding
author must be affiliated with the chosen institute. Please either amend your
manuscript to change the affiliation or corresponding author, or email us at plosone@plos.org with a request to remove this
option.

8. Please ensure that you refer to Figures 1 and 3 in your text as, if accepted,
production will need this reference to link the reader to the figure.

Reviewer's Responses to Questions

**Comments to the Author**

1. Is the manuscript technically sound, and do the data support the conclusions?

Partly

2. Has the statistical analysis been performed
appropriately and rigorously? 

Yes

4. Is the manuscript presented in an intelligible
fashion and written in standard English?

Reviewer #1: No

5. Review Comments to the Author

Reviewer #1: The study is interesting, and the results are useful for diversity
related research. However, the manuscript needs to be thoroughly checked for
language including grammar. Detailed review comments are attached.

Reviewer #2: Dear Authors

You get a minor revision, make a revision immediately, pay attention to the marked
reviews, the introduction is quite clear but there are some things that need to be
improved, the methods, data processing and others are quite good

Thank you

See reviewer comments as an attachment file.

Review_2021.docx
Paper-01.docx
---

## [Author Response · Author response to Decision Letter 0]

24 Jan 2022

General: The whole paper was revisited and the help of the expert/native speaker was
also incorporated. Additionally, some technical quality improvements are also
considered. 

Reviewer #1: The study is interesting, and the results are useful for diversity
related research. However, the manuscript needs to be thoroughly checked for
language including grammar. Detailed review comments are attached.

 Suggestion accepted and incorporated . Thanks for your appreciation, the whole paper
was revisited and the suggestions were incorporated 

Reviewer #2: You get a minor revision, make a revision immediately, pay attention to
the marked reviews, the introduction is quite clear but there are some things that
need to be improved, the methods, data processing and others are quite good

 Suggestion accepted and incorporated. Thanks for your appreciation, the whole paper
was revisited and the suggestions were incorporated

---

## [Decision Letter · Decision Letter 1]

21 Jun 2022

Diversity Impact on Organisational Performance:

Moderating and Mediating Role of Diversity Beliefs and Leadership Expertise in
Pakistan

PONE-D-21-09151R1

Dear Dr.
Shahryar
Sorooshian,

We’re pleased to inform you that your manuscript has been judged scientifically
suitable for publication and will be formally accepted for publication once it meets
all outstanding technical requirements.

Kind regards,

María del Carmen Valls Martínez, Ph.D.

Academic Editor

PLOS ONE

Reviewers' comments:

Reviewer's Responses to Questions

**Comments to the Author**

1. If the authors have adequately addressed your comments raised in a previous round
of review and you feel that this manuscript is now acceptable for publication, you
may indicate that here to bypass the “Comments to the Author” section, enter your
conflict of interest statement in the “Confidential to Editor” section, and submit
your "Accept" recommendation.

Reviewer #2: All comments have been addressed

Reviewer #3: All comments have been addressed

2. Is the manuscript technically sound, and do the data
support the conclusions?

Reviewer #2: Yes

Reviewer #3: Yes

3. Has the statistical analysis been performed
appropriately and rigorously? 

Reviewer #2: Yes

Reviewer #3: Yes

4. Have the authors made all data underlying the
findings in their manuscript fully available?

Reviewer #2: Yes

Reviewer #3: Yes

5. Is the manuscript presented in an intelligible
fashion and written in standard English?

Reviewer #2: Yes

Reviewer #3: Yes

6. Review Comments to the Author

Reviewer #2: Based on the results of the previous reviewer's comments, which
generally have to be revised by the author. Well done.

Reviewer #3: The study is very interesting, and the overall results are useful for
related research areas. Therefore, I recommend to publish this article in its
current form.

7. PLOS authors have the option to publish the peer
review history of their article (what does this mean?). If published, this will
include your full peer review and any attached files.

If you choose “no”, your identity will remain anonymous but your review may still be
made public.

**Do you want your identity to be public for this peer review?** For
information about this choice, including consent withdrawal, please see our
Privacy Policy.

Reviewer #2: No

Reviewer #3: **Yes: **Dr. Asad Ali

---

## [Editor Report · Acceptance letter]

15 Jul 2022

PONE-D-21-09151R1 

Diversity Impact on Organizational performance: Moderating and Mediating role of
Diversity Beliefs and Leadership Expertise 

Dear Dr. Sorooshian:

I'm pleased to inform you that your manuscript has been deemed suitable for
publication in PLOS ONE. Congratulations! Your manuscript is now with our production
department. 

Kind regards, 

on behalf of

Dr. María del Carmen Valls Martínez 

Academic Editor

PLOS ONE